# Influence of Prostate Cancer on Thulium Vapoenucleation of the Prostate—A Multicentre Analysis

**DOI:** 10.3390/jcm12031174

**Published:** 2023-02-01

**Authors:** Tobias Lamersdorf, Christopher Netsch, Benedikt Becker, Christian Wülfing, Petra Anheuser, Oliver Engel, Andreas J. Gross, Clemens Mathias Rosenbaum

**Affiliations:** 1Department of Urology, Asklepios Hospital Barmbek, 22307 Hamburg, Germany; 2Department of Urology, Asklepios Hospital Altona, 22763 Hamburg, Germany; 3Department of Urology, Asklepios Hospital Wandsbek, 22043 Hamburg, Germany; 4Department of Urology, Asklepios Hospital Harburg, 21075 Hamburg, Germany

**Keywords:** benign prostate hyperplasia, enucleation, endourology, laser, prostate neoplasm

## Abstract

*Purpose:* Prostate cancer (PCa) and benign prostatic hyperplasia (BPH) are common in elderly men. Data on the laser-based surgery known as thulium vapoenucleation of the prostate (ThuVEP) in PCa patients are rare. Our objective was to analyse the feasibility, safety and functional outcome of ThuVEP in patients with lower urinary tract symptoms (LUTS) and PCa. *Methods:* Multicentre study, including 1256 men who underwent ThuVEP for LUTS. Maximum urinary flow rate (Qmax) and post-void residual volume (PVR) were measured perioperatively. The International Prostate Symptome Score (IPSS) was measured perioperatively and at follow-up (FU). Perioperative complications were captured. Reoperation rate was captured at FU. *Results:* Of 994 men with complete data, 286 (28.8%) patients had PCa. The most common Gleason score was 3 + 3 in 142 patients (49.7%). Most common was low-risk PCa (141 pts; 49.3%). PCa patients were older, had smaller prostates and had higher prostate-specific antigen (PSA) values (all *p* < 0.001). Comparing non-PCa and PCa patients, no differences occurred perioperatively. IPSS, quality of life and PVR decreased (all *p* < 0.001) and Qmax improved (*p* < 0.001) in both groups. Reoperation rates did not differ. The results of low- vs. intermediate-/high-risk PCa patients were comparable. *Conclusion:* ThuVEP is a safe and long-lasting treatment option for patients with LUTS with or without PCa. No differences occurred when comparing low- to intermediate-/high-risk PCa patients.

## 1. Introduction

Benign prostatic obstruction (BPO) is a common disease of elderly men. More than 50% of men older than 70 years suffer from moderate to severe lower urinary tract symptoms (LUTS) [1]. After failed conservative management many patients need surgical treatment [2]. In Germany, approximately 75,000 patients undergo BPO surgery per year [3]. 

At the same time, prostate cancer (PCa) represents the most common cancer of men [4]. The diagnosis of PCa as incidental PCa is a common histologic finding in patients undergoing BPO surgery. Moreover, patients with a known PCa are likely to suffer BPO symptoms that might lead to BPO surgery [5]. 

Data about perioperative and functional outcomes of patients with PCa who undergo desobstruction of the prostate are rare. TURP was described as a safe treatment in patients with PCa [6]. Crain et al. compared the palliative TURP in a small group of 19 patients with PCa to patients undergoing TURP with BPO. Besides TURP, laser-based surgeries including thulium vapoenucleation of the prostate (ThuVEP), photoselective vaporization of the prostate (PVP) or holmium laser enucleation of the prostate (HoLEP) were increasingly performed over the last two decades [7]. However, there are few studies on the feasibility of laser-based surgery in patients with PCa. Chen et al. described PVP in 39 PCa patients with no control group [8,9]. Data on HoLEP in patients with known PCa are described by Becker et al. [10]. They reported on the feasibility of HoLEP in a small group of 62 patients with biopsy-proven PCa. Elkoushy et al. compared 70 patients with incidental PCa (iPCa) after HoLEP to a large group of BPH patients [11]. Both authors describe comparable functional outcomes. 

To the best of our knowledge, there are no publications on PCa in ThuVEP patients. Moreover, the quoted literature only analyses small cohorts of patients and mostly deals with single-centre data. Therefore, we aimed to analyse a large cohort of patients undergoing ThuVEP in three tertiary care hospitals. Our hypothesis is that ThuVEP is a feasible and safe procedure in PCa patients.

## 2. Material and Methods

After receiving institutional review board approval, we conducted this multicentre and non-selective consecutive data collection involving all men who underwent ThuVEP for symptomatic BPO between January 2017 and January 2020 in three urological departments. Five surgeons performed ThuVEP during the above-mentioned period. All participating surgeons had already performed more than 200 ThuVEPs. ThuVEP was always performed completely. No tunneling or only enucleating the middle lobe was been performed. Pre- and post-operative demographics, perioperative course, and patients’ characteristics such as age, American Society of Anesthesiologists (ASA) score, body mass index (BMI), prostate-specific antigen (PSA) and prostate volume were recorded. Prostate volume measured by TRUS was calculated using prolate ellipse volume ([cm^3^] = (length × width × height) × π/6). We analyzed the International Prostate Symptome Score (IPSS), quality of life (QoL), maximum urinary flow rate (Qmax) and post-void residual volume (PVR). We did not retrieve information about self-reported symptoms of inflammation indicating possible prostatitis. Postoperative complications were reported according to the Clavien–Dindo System. 

A yearly follow-up was conducted by mail capturing IPSS, QoL and reoperation rate. 

We assessed pathologic outcome retrospectively. Known presence of PCa prior to ThuVEP or data about prostate biopsy were not captured in our database. Therefore, we were not able to differentiate between already-diagnosed PCa and first PCa diagnosis by surgery (i.e., incidental prostate cancer). In the participating centres, digital rectal examination, transrectal ultrasound of the prostate and PSA testing are part of the preoperative workup in ThuVEP patients. If any suspicious findings were detected, further PCa diagnostics were initiated in accordance with the European guidelines [12]. 

We compared demographics, morbidity and peri- and postoperative course of the PCa group in comparison to the non-PCa group in order to identify differences. We subdivided PCa patients into three groups: low-risk as preoperative PSA ≤ 10 ng/mL and Gleason score ≤ 6 in final pathology; intermediate-risk as preoperative PSA 10–20 ng/mL and/or Gleason score 7 in final pathology; high-risk as preoperative PSA ≥ 20 ng/mL and/or Gleason score ≥ 8 in final pathology. We compared demographics, morbidity and peri- and postoperative course of the low- to the intermediate-/high-risk groups in order to identify significant differences. Further treatment of PCa or oncologic outcome was not captured in our database. 

We performed the Kolmogorov–Smirnov test to analyze the normality of the distribution. Comparisons between groups were performed using *t*-test, Pearson’s Chi-squared test and Mann–Whitney U-test according to the distribution. The significance level was set at α < 0.05.

## 3. Results

A total of 1256 patients underwent ThuVEP at three institutions. Of those, 262 patients were excluded due to incomplete data or incomplete pathology. Of the 994 included patients, PCa as final pathology occurred in 286 patients (28.8%). The Gleason score distribution was 3 + 3 in 142 (49.7%), 3 + 4 in 100 (35.0%), 4 + 3 in eight (2.8%), 4 + 4 in four (1.7%), 4 + 5 in 17 (5.9%) and 5 + 5 in six (2.1%) patients. Risk classification showed 141 low-risk patients (49.3%), 110 intermediate-risk patients (38.5%) and 25 high-risk patients (8.7%). 

Baseline characteristics of PCa and non-PCa patients are displayed in Table 1. PCa patients were significantly older, had a smaller prostate volume and had a higher PSA value (all *p* < 0.001). Preoperatively, Qmax was significantly weaker in PCa patients (*p* = 0.011) compared to non-PCa patients. Around one third of patients in both groups had a transurethral catheter preoperatively (non-PCa patients vs. PCa patients 36.2% vs. 35.7%, respectively; *p* = 0.14). 

Table 2 displays the intra- and post-operative outcomes. Enucleated prostate volume was significantly lower in PCa patients. The percentage of enucleated volume compared to the preoperative measured volume of the prostate was almost the same (69% in non-PCa patients and 66% in PCa patients). All other intraoperative parameters were comparable. Early functional parameters, measured after catheter removal usually on the second or third post-operative day, were comparable. IPSS (9.8 vs. 9.3; *p* = 0.30) and QoL (1.9 vs. 1.8; *p* = 0.39) did not differ significantly. Qmax (19.2 mL/s vs. 18.1 mL/s; *p* = 0.15) and PVR (48.1 vs. 51.0; *p* = 0.52) did not differ significantly. Numbers of patients with a transurethral catheter at discharge were comparable (non-PCa patients vs. PCa patients 5.6% vs. 8.4%, respectively; *p* = 0.055).

The mean pre- and post-operative IPSS, QoL and PVR decreased significantly in both groups (all *p* < 0.001). The mean Qmax improved significantly in both groups (*p* < 0.001).

Table 3 shows the first- and second-year follow-up data of patients stratified by final pathology. In all, 505 patients completed the first-year follow-up and 249 patients completed second-year follow-up. IPSS and QoL did not differ among groups. The rate of reintervention did not differ.

Comparing low-risk PCa patients (n = 141) to intermediate- and high-risk PCa patients (n = 135), there were no significant differences in the baseline characteristics age (71.8 vs. 73.0 years; *p* = 0.22), BMI (26.5 vs. 26.8 kg/m^2^; *p* = 0.70), measured prostate volume (65.8 vs. 61.7 cc; *p* = 0.38) and PSA (6.2 vs. 8.7 ng/mL; *p* = 0.61). Moreover, no difference occurred when comparing the preoperative functional results IPSS (18.4 vs. 18.5; *p* = 0.89), QoL (4.0 vs. 4.3; *p* = 0.15), Qmax (9.0 vs. 9.4 mL/s; *p* = 0.68) and PVR (161.0 vs. 131.3 mL; *p* = 0.64). 

Comparing intra- and perioperative results as rates of perforation of the prostate capsule, bleeding, blood transfusion or reoperation during hospital stay, there were no differences among low-risk PCa patients and intermediate- or high-risk PCa patients. IPSS (8.6 vs. 9.8; *p* = 0.17) at discharge did not differ significantly whereas QoL (1.6 vs. 2.0, *p* = 0.047) did. Qmax at discharge differed between low- and intermediate/high-risk PCA patients (19.9 vs. 16.8 mL/s; *p* = 0.015), but PVR did not differ (51.7 vs. 50.2 mL; *p* = 0.85). These differences did not occur at first- and second-year follow-ups. Further on, the reoperation rate was comparable at first- and second-year follow-ups among the groups.

## 4. Discussion

Due to the increasing age of patients undergoing surgical BPO treatment, the presence of PCa in final pathology increases [13]. To the best of our knowledge no study exists that compares a larger cohort of patients with PCa and without PCa undergoing laser-based BPO surgery. Neither perioperative risks nor mid- to long-term outcomes have been sufficiently investigated in this group of patients. 

ThuVEP provides a long-lasting, significant improvement in functional outcome in patients with and without PCa. Early outcomes were comparable. All measured functional results (IPSS, Qmax, QoL, PVR) improved significantly in both groups. 

Even more importantly, our study shows almost no differences in mid-term outcome measures comparing PCa patients with non-PCa patients. When comparing the functional outcome at one and two years after surgery, IPSS and QoL domains display high and long-lasting therapy success. The reoperation rate is comparable during the follow-up time. The long-lasting improvement in functional outcome independent of final pathology may surprise since the anatomical differences in PCa patients are reported by many surgeons. Moreover, the idea that PCa patients are more likely to progress and cause obstruction again could not be proven in our cohort. A possible explanation is the percentage of enucleated prostate volume, which was almost the same in both groups and allows regrowth of the prostate to a certain extent without causing LUTS. Our results do not reflect the findings of Crain and colleagues. Here, the functional outcome after TUR-P in PCa patients was worse. Inferior results after palliative TUR-P are probably caused by a different operation technique in palliative settings. Crain et al. described resecting only a “channel” to avoid or minimise severe adverse events [6]. Our study shows that it is not necessary to treat PCa patients differently to avoid adverse events. They are not more frequent in PCa patients undergoing ThuVEP. 

Comparable mid-term results after HoLEP were also shown by Elkoushy et al., underlining the feasibility of laser enucleation in patients with PCa [11]. They showed a very small risk of progression of the PCa in their follow-up of 48 months. 

Perioperative course did not differ among groups. Peri- and postoperative complication rates were low in both groups and comparable to large groups of BPH patients undergoing ThuVEP [14], highlighting the safety of ThuVEP in PCa patients. Our current data show low rates of relevant bleeding translating into low blood transfusion and reoperation rates. 

In our cohort PCa was apparent in 286 patients (28.8%). The most common Gleason score distribution was 3 + 3 (142 patients; 49.7% of all PCa patients). 

These detection rates are higher than given data in the literature. Even though rates of incidentally detected PCa have decreased in the past years [15], in TUR-P patients, rates were only about 5–10% [16,17]. In HoLEP cohorts, detection rates were higher with 6.8–23% [18,19,20]. The higher detection rate of PCa in our study may be due to the fact that PCa cases in our cohort were not only incidental PCa but also known PCa patients who underwent ThuVEP due to LUTS caused by PCa. We retrospectively assessed pathological data and we were not able to evaluate PCa status prior to ThuVEP. Therefore, we were not able to differentiate between preoperatively known or unknown PCa. Another possible explanation of our high detection rate of PCa might be the high percentage of enucleated prostate tissue. This theory was presented by Rosenhammer et al. [20] where resected/enucleated tissue and incidental prostate cancer (i-PCa) detection rate in HoLEP and TURP patients were compared. Significantly higher rates in HoLEP patients were found. Our data of almost 1000 patients underline their findings. Moreover, the rate of enucleated prostate tissue was higher in our study (69%), compared to Carmignani et al. [21] (24%) who showed an i-PCa rate of only 2.5% in ThuVEP patients. 

When comparing the short- and mid-term outcomes between low- and intermediate/high-risk PCA patients, there were only minor differences in Qmax, IPSS and QoL. All parameters were slightly better in low-risk PCa patients. 

We present a large cohort of patients in a multicentre data collection. However, our study has some limitations. First, there was a high dropout rate during the follow-up. This possibly leads to biased data. Second, pathological outcomes were assessed retrospectively. We were not able to capture preoperative biopsy data or PCa therapy sequences prior to ThuVEP. Hence, we were not able to differentiate between preoperatively known or unknown PCa. Additionally, preoperative oncologic data such as PSA density were not captured. Furthermore, we did not capture signs of prostatitis, possibly influencing symptoms and outcomes [22]. Third, missing follow-up of PCa treatment leads to a potential bias. The functional outcome could be influenced by the treatment. 

However, we are the first to demonstrate the feasibility and safety of ThuVEP for prostatic obstruction in patients with and without PCa, analysing the largest cohort of patients published so far. 

## 5. Conclusions

ThuVEP for prostatic obstruction is a feasible and safe procedure not only in BPO but also in PCa patients. Perioperative and mid-term functional outcome is comparable among PCA and non-PCa patients after ThuVEP. 

## Figures and Tables

**Table 1 jcm-12-01174-t001:** Baseline and preoperative characteristics of patients stratified by final pathology.

	All	No PCa	PCa	*p*-Value
Number of patients; n (%)	994 (100)	708 (71.2)	286 (28.8)	-
Age (years); mean (SD)	71.3 (±8.1)	70.6 (±8.1)	72.9 (±7.9)	<0.001 *
ASA score; mean (SD)	2.5 (±0.7)	2.5 (±0.7)	2.5 (±0.7)	0.8 +
BMI (kg/m^2^); mean (SD)	27.0 (±4.2)	27.1 (±4.3)	26.6 (±3.7)	0.19 *
Measured prostate volume (cc); mean (SD)	72.5 (±41.6)	76.4 (±42.7)	63.1 (±37.1)	<0.001 *
PSA preop (ng/mL); mean (SD)	7.9 (±22.9)	5.9 (±7.5)	12.9 (±40.3)	<0.001 *
IPSS preop (points); mean (SD)	18.9 (±7.2)	18.9 (±7.1)	19.0 (±7.6)	0.8 *
QoL preop (points); mean (SD)	4.3 (±1.4)	4.3 (±1.4)	4.2 (±1.5)	0.08 *
Qmax preop (mL/s); mean (SD)	10.4 (±6.1)	11.0 (±6.6)	9.2 (±4.8)	0.011 *
PVR preop (mL); mean (SD)	174.8 (±187.1)	174.1 (±200.0)	176.6 (±154.7)	0.8 *

* *t*-test; + Mann–Whitney U-test.

**Table 2 jcm-12-01174-t002:** Intra- and early postoperative characteristics of patients stratified by final pathology.

	All	No PCa	PCa	*p*-Value
Number of patients; n (%)	994 (100)	708 (71.2)	286 (28.8)	-
Perforation of prostate capsule; n (%)	134 (14.3)	100 (15.2)	34 (12.4)	0.27 ^#^
Relevant bleeding intraoperatively; n (%)	58 (6.2)	43 (6.5)	15 (5.5)	0.54 ^#^
Resected prostate volume (cc); mean (SD)	50.2 (±37.7)	53.5 (±37.4)	42.3 (±37.4)	<0.001 *
Blood transfusion rate; n (%)	12 (1.2)	10 (1.4)	2 (0.7)	0.10 ^#^
Reoperation rate; n (%)	60 (6.0)	46 (6.5)	14 (4.9)	0.12 ^#^

Perforation of prostate capsule as reported by the surgeon. Relevant bleeding as reported by the surgeon. Blood transfusion rate: No patient received more than one blood transfusion. Reoperation during hospital stay. ^#^ Chi-square test; * *t*-test.

**Table 3 jcm-12-01174-t003:** First- and second-year follow-up data of patients stratified by final pathology.

		All	No PCa	PCa	*p*-Value
First-year follow-up					
IPSS (points); mean (SD)		7.1 (±6.8)	6.9 (±6.8)	7.5 (±6.7)	0.37 **
QoL (points); mean (SD)		1.8 (±2.1)	1.8 (±2.2)	1.9 (±1.7)	0.38 **
BOO—Reoperation rate +					
	No reoperation; n (%)	443 (87.7)	311 (88.1)	130 (86.7)	0.38 *
	Reoperation; n (%)	62 (12.3)	42 (11.9)	20 (13.3)	
Second-year follow-up					
IPSS (points); mean (SD)		7.0 (±6.6)	6.8 (±6.4)	7.9 (±7.2)	0.29 *
QoL (points); mean (SD)		1.8 (±1.6)	1.7 (±1.6)	1.9 (±1.7)	0.45 *
BOO—Reoperation rate ^#^					
	No reoperation; n (%)	242 (96.4)	191 (97.0)	51 (94.4)	0.30 *
	Reoperation; n (%)	9 (3.6)	6 (3.0)	3 (5.6)	

+ Reoperation after discharge within first-year FU. ^#^ Reoperation within second-year FU. * Chi-square test; ** *t*-test.

## Data Availability

The data that support the findings of this study are not publicly available but are available from the corresponding author [CMR] upon reasonable request.

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
