# Peer review of "Influence of Prostate Cancer on Thulium Vapoenucleation of the Prostate—A Multicentre Analysis"

_jcm, 2023, doi:10.3390/jcm12031174_

Round 1

Reviewer 1 Report

The paper is well written the designa and results are clear.

There are however some information which shold be considered.

1) In the discussion the authors report: "Due to the increasing age of patients undergoing surgical BPO treatment, presence of PCa in final pathology increases".There lacks a reference. I suggest to add this recent reference. Ferraro, S.et al. Definition of Outcome-Based Prostate-Specific Antigen (PSA) Thresholds for Advanced Prostate Cancer Risk Prediction. Cancers 2021, 13, 3381. 

In this paper it is reported that older patients are at higher risk for PCa.

2)There is another evidence which may be relevant retrieved from  the previous paper concerning the presence of prostate inflammation(Histological Evidence). Ferraro et al. have reported that glandular inflammation is prevalent in low risk Prostate cancer patients, representing the PCa cases evaluated in this study .

The authors have not retrieved information about self-reported sympthoms of inflammation, which is relevant to report in LUTS(this may be considered as a limit). Look at this reference:Ficarra V. et alThe role of inflammation in lower urinary tract symptoms (LUTS) due to benign prostatic hyperplasia (BPH) and its potential impact on medical therapy.

Author Response

The paper is well written the design and results are clear. There are however some information which should be considered.

1) In the discussion the authors report: "Due to the increasing age of patients undergoing surgical BPO treatment, presence of PCa in final pathology increases". There lacks a reference. I suggest to add this recent reference. Ferraro, S.et al. Definition of Outcome-Based Prostate-Specific Antigen (PSA) Thresholds for Advanced Prostate Cancer Risk Prediction. Cancers 2021, 13, 3381. 

In this paper it is reported that older patients are at higher risk for PCa.

2)There is another evidence which may be relevant retrieved from the previous paper concerning the presence of prostate inflammation (Histological Evidence). Ferraro et al. have reported that glandular inflammation is prevalent in low-risk Prostate cancer patients, representing the PCa cases evaluated in this study.

Answer: We want to thank the reviewer for this important remark. We added the reference as suggested. We agree that it fits very well.

The authors have not retrieved information about self-reported sympthoms of inflammation, which is relevant to report in LUTS (this may be considered as a limit). Look at this reference: Ficarra V. et al. The role of inflammation in lower urinary tract symptoms (LUTS) due to benign prostatic hyperplasia (BPH) and its potential impact on medical therapy.

Answer: We thank the reviewer for this relevant comment. We agree, that symptoms, possibly hinting towards inflammation are an important part of LUTS. However, we did not capture them. We added in Material and Methods:

“We did not retrieved information about self-reported symptoms of inflammation indicating possible prostatitis.”

Additionally, we added in Discussion (in the limitations part):

“Furthermore, we did not capture signs of prostatitis, possibly influencing symptoms and outcomes”

We added the above-mentioned reference.

Reviewer 2 Report

The author provided evidence related to the laser-based surgery as thulium vapoenucleation of the prostate (ThuVEP) in PCa patient.

Major

1. Since ThuVEP in Pca may not be recommended in Pca patients. Could you describe how application of the procedure is determined.  Or do you adapt the procedure intentionally to the Pca patients? 

2. Some report mentioned metastasis after ThuVEP in Pca patients. Could you describe the presence (rate) of metastasis after procedure among Pca patients. 

3. The recovery of IPSS after 1st and second year may be due to the response to the androgen deprivation therapy? Please add the table describe the treatment of Pca patients 

Author Response

The author provided evidence related to the laser-based surgery as thulium vapoenucleation of the prostate (ThuVEP) in PCa patient.

1. Since ThuVEP in Pca may not be recommended in Pca patients. Could you describe how application of the procedure is determined. Or do you adapt the procedure intentionally to the Pca patients?

Answer: We thank the reviewer for this comment. However, to the best of our knowledge, there is no guideline saying ThuVEP must not be performed in PCa patients if they suffer of LUTS and are not willing to undergo radical prostatectomy. Of course, patients are evaluated preoperatively due to guideline recommendations. We discussed this in Material and Methods:

“In the participating centers, digital rectal examination, transrectal ultrasound of the prostate and PSA testing are part of the preoperative workup in ThuVEP patients. If any suspicious findings were detected, further PCa diagnostics were initiated in accordance with the European guidelines” 

Stil, adhearing to those recommendations may possibly lead to missing out some PCa cases. We discussed this in Introduction:

“At the same time, Prostate cancer (PCa) represents the most common cancer of men. The diagnosis of PCa as incidental PCa is a common histologic finding in patients undergoing BPO surgery.”

Irrespectively of an existing or known PCa, we always perform an enucleation of the whole prostate. We elaborated this in Material and Methods:

“ThuVEP was always performed completely. No tunneling or only enucleating the middle lobe has been performed.”

Moreover, we discussed this in Discussion:

“The long-lasting improvement in functional outcome, independent of final pathology may surprise since the anatomical differences in PCa patients are reported by many surgeons. Moreover, the idea that PCa patients are more likely to progress, and cause obstruction again could not be proven in our cohort. A possible explanation is the percentage of enucleated prostate volume, which is almost the same in both groups and allows regrowth of the prostate to a certain extent without causing LUTS. Our results do not reflect the findings of Crain and colleagues. Here, functional outcome after TUR-P in PCa patients was worse. Inferior results after palliative TUR-P are probably caused by a different operation technique in palliative settings. Crain et al. described to resect only a “channel” to avoid or minimise severe adverse events”

2. Some report mentioned metastasis after ThuVEP in Pca patients. Could you describe the presence (rate) of metastasis after procedure among Pca patients. 

We want to thank the reviewer for this interesting thought. We do not believe that there is a higher rate of metastases in PCa patients after ThuVEP. Especially in the light of high rates of Gleason Score 3+3 patients and high rates of low risk PCa patients as in our cohort. However, we are not able to prove this statement. We present a retrospective analysis of prospectively collected, functional data. Initial focus of our data set was not oncological at the beginning. We are aware of this problem. In discussion we stated:

“Second, pathological outcomes were assessed retrospectively. We were not able to capture preoperative biopsy data or PCa therapy sequences prior to ThuVEP.”

Moreover, we tried to stress this issue in Material and Methods. We added:

“Further treatment of PCa or oncologic outcome was not captured in our database.”  

3. The recovery of IPSS after 1st and second year may be due to the response to the androgen deprivation therapy? Please add the table describe the treatment of Pca patients 

We believe, the reviewer emphasizes an important issue. Unfortunately, we did not capture further oncologic therapy or outcome in our data base. We added this in Material and Methods:

“Further treatment of PCa or oncologic outcome was not captured in our database”

However, when comparing low- to intermediate-/high-risk-PCa patients we did not find any difference:

“Comparing low-risk-PCa-patients (n=141) to intermediate- and high-risk-PCa-patients (n=135), there were no significant differences in baseline characteristics as age (71.8 vs. 73.0 years; p=0.22), BMI (26.5 vs. 26.8kg/m²; p=0.70), measured prostate volume (65.8 vs. 61.7cc; p=0.38) and PSA (6.2 vs. 8.7ng/ml; p=0.61). Moreover, no difference occurred when comparing preoperative functional results as IPSS (18.4 vs. 18.5; p=0.89), QoL (4.0 vs. 4.3; p=0.15), Qmax (9.0 vs. 9.4ml/sec; p=0.68) and PVR (161.0 vs. 131.3ml; p=0.64).”

Therefore, we believe, there is no relevant impact of oncologic therapy on functional outcome in our cohort. However, we addressed the issue in Discussion:

“Second, pathological outcomes were assessed retrospectively. We were not able to capture preoperative biopsy data or PCa therapy sequences prior to ThuVEP. […] Third, missing follow-up of PCa treatment leads to a potential bias. Functional outcome could be influenced by the treatment.”

Reviewer 3 Report

Authors should be congratulated for their work. ThuVEP is a challenging technique that represents a valid option in patients treated with anticoagulants or with bleeding disorders. The manuscript is well written but several points warrant a mention:

- Which was the risk classification system used for PCa patients? The criteria reported are missing of T-stage. Before surgery, did the prostate checked with digital rectal examination? (PMID: 36165471)

- Why did the Authors not use a propensity score-matched analysis? The two samples are numerically different.

- Are data available on PSA density? Several studies (PMID: 29857868) considered PSAd as a preliminary tool for detection of PCa.

- When the PCa diagnosis was established, did the PCa patients undergo disease-specific management? Was the prostatectomy performed? Any data are available on final histology? And on the vanishing tumor phenomenon?

-Before the surgery, had patients had urethral catheters? Or experienced an acute urinary retention?

-Did patients use drugs to improve the LUTS? How changed this therapy after surgery?

Author Response

Authors should be congratulated for their work. ThuVEP is a challenging technique that represents a valid option in patients treated with anticoagulants or with bleeding disorders. The manuscript is well written but several points warrant a mention:

We want to thank the reviewer for the positive feedback. We tried to point out, that ThuVEP is not only helpful in patients treated with anticoagulants or with bleeding disorders but can be done in Prostate Cancer patients as well.

- Which was the risk classification system used for PCa patients? The criteria reported are missing of T-stage. Before surgery, did the prostate checked with digital rectal examination? (PMID: 36165471)

We tried to adapt the D’Amico risk classification to the parameters we had. The reviewer is right: We did not have a reliable T-stage. Therefore, we avoided to use the term “D’Amico low-/intermediate-/high-risk”. We described the risk stratification in Material and Methods:

“We subdivided PCa patients into three groups: Low-risk as preoperative PSA ≤10 ng/ml and Gleason score ≤6 in final pathology; Intermediate-risk as preoperative PSA 10-20 ng/ml and/or Gleason score 7 in final pathology; High-risk as preoperative PSA ≥20 ng/ml and/or Gleason score ≥ 8 in final pathology.”

Moreover, we described our preoperative investigations in every patient potentially undergoing ThuVEP in our departments:

“In the participating centers, digital rectal examination, transrectal ultrasound of the prostate and PSA testing are part of the preoperative workup in ThuVEP patients. If any suspicious findings were detected, further PCa diagnostics were initiated in accordance with the European guidelines.”

- Why did the Authors not use a propensity score-matched analysis? The two samples are numerically different.

We believe, propensity score-matching would lead to increased model imbalance and inefficiency in our cohort. We believe, the matching methods we used are more valid in the cohort we investigated. We think, that matching on observed variables may unleash bias due to dormant unobserved confounders. Moreover, we believe, the confounder “further therapy”, which we did not capture, is too relevant. As a propensity score-matched analysis only accounts for observed covariates, we believe this factor would relevantly affect the results and would have led to hidden bias. Therefore, we decided not to use a propensity score-matched analysis

- Are data available on PSA density? Several studies (PMID: 29857868) considered PSAd as a preliminary tool for detection of PCa.

The reviewer points out an important option to judge on potential PCa in BPH patients. We pointed out in Material and Methods:

“In the participating centers, digital rectal examination, transrectal ultrasound of the prostate and PSA testing are part of the preoperative workup in ThuVEP patients. If any suspicious findings were detected, further PCa diagnostics were initiated in accordance with the European guidelines.”

However, we could not rely on solid PSA density data. We tried to emphasize the issue of an oncologic related analysis in a functional data base. We added in Discussion:

“Second, pathological outcomes were assessed retrospectively. We were not able to capture preoperative biopsy data or PCa therapy sequences prior to ThuVEP. Hence, we were not able to differentiate between preoperatively known or unknown PCa. Additionally, preoperative oncologic data such as PSA density were not captured.”  

- When the PCa diagnosis was established, did the PCa patients undergo disease-specific management? Was the prostatectomy performed? Any data are available on final histology? And on the vanishing tumor phenomenon?

We thank the reviewer for the important remark. However, as we pointed out in Material and Methods:

“Further treatment of PCa or oncologic outcome was not captured in our database. “

We tried to further analyse PCa treatment. However, data was incomplete, most likely due to the high drop out rate. We addressed this in Discussion:

“First, there is a high dropout rate during the follow-up. This possibly leads to biased data.”

-Before the surgery, had patients had urethral catheters? Or experienced an acute urinary retention?

In Results we wrote:

“Around one third of patients in both groups had a transurethral catheter preoperatively (non-PCa patients vs. PCa patients 36.2% vs. 35.7%; p=0.14).”

-Did patients use drugs to improve the LUTS? How changed this therapy after surgery?

The reviewer points out an important issue. Preoperative medical therapy of LUTS could have an impact on surgery and outcome. Additionally, some medication leads to reduced prostate volume and PSA values. We tried to analyze medical therapy prior to surgery. However, data seemed incomplete. Therefore, we decided not to add this information in our analysis.

Reviewer 4 Report

Review for the Journal of Clinical Medicine (Comments to the Author)

For the paper entitled: „Influence of Prostate Cancer on Thulium Vapoenucleation of the Prostate – A Multicentre Analysis“ are aspects that should be considered.

The authors studied a large population, which gives the work high priority.

„Known presence of PCa prior to 79 ThuVEP or data about prostate biopsy has not been captured in our database. Therefore, 80 we were not able to differentiate between already diagnosed PCa and first PCa diagnosis 81 by surgery (i.e. incidental prostate cancer).“ is a limitation of the study that reduces the meaningfulness. It is good that the authors explain the limitations in detail and also included them in the discussion.

It is positive that the authors, when considering the results, divided the patients with a Gleason score of 7 into the two clinically separate subgroups 3+4 = 7a and 4+3 = 7b.

In Line 198 the authors discuss the results of Rosenhammer et al. (Literatur No. 19) and use the abbreviation iPCa. The authors should define the abbreviation like „incidental prostate cancer (iPCa)“.

In addition, Qmax in the abstract and the manuscript should be defined.

Author Response

For the paper entitled: „Influence of Prostate Cancer on Thulium Vapoenucleation of the Prostate – A Multicentre Analysis“ are aspects that should be considered. The authors studied a large population, which gives the work high priority.

We thank the reviewer for this positive feedback. We believe, the large population is one of the major strengths. Of our analysis.

„Known presence of PCa prior to ThuVEP or data about prostate biopsy has not been captured in our database. Therefore, we were not able to differentiate between already diagnosed PCa and first PCa diagnosis by surgery (i.e. incidental prostate cancer).“ is a limitation of the study that reduces the meaningfulness. It is good that the authors explain the limitations in detail and also included them in the discussion.

The reviewer points out an important issue. We are aware of this drawback of our data set. In reply to other reviewer concerns, we added in Discussion:

“First, there is a high dropout rate during the follow-up. This possibly leads to biased data. Second, pathological outcomes were assessed retrospectively. We were not able to capture preoperative biopsy data or PCa therapy sequences prior to ThuVEP. Hence, we were not able to differentiate between preoperatively known or unknown PCa. Additionally, preoperative oncologic data such as PSA density were not captured.” 

It is positive that the authors, when considering the results, divided the patients with a Gleason score of 7 into the two clinically separate subgroups 3+4 = 7a and 4+3 = 7b.

We agree with the reviewer. We believe (as it is backed by many publications) that Gleason score 3+4 = 7a and 4+3 = 7b make a relevant difference.

In Line 198 the authors discuss the results of Rosenhammer et al. (Literatur No. 19) and use the abbreviation iPCa. The authors should define the abbreviation like „incidental prostate cancer (iPCa)“.

We want to thank the reviewer for this important remark. We defined iPCA as incidental prostate cancer in discussion. After that, we moved on using the abbreviation.

In addition, Qmax in the abstract and the manuscript should be defined.

We thank the reviewer for this important note. We defined Qmax in Abstract and in Material and Methods as maximum urinary flow rate (Qmax).

Round 2

Reviewer 2 Report

The author has made the appropriate corrections. No further request. 

Reviewer 3 Report

Authors should be congratulated. They improved the quality of the manuscript which is now suitable for publication